# Item response theory model highlighting rating scale of a rubric and rater–rubric interaction in objective structured clinical examination

**Masaki Uto**[1]*, **Jun Tsuruta**[2], **Kouji Araki**[3], **Maomi Ueno**[1]

**1** Department of Computer and Network Engineering, The University of Electro-Communications, Chofu, Tokyo, Japan, **2** Institute of Education, Tokyo Medical and Dental University, Bunkyo-ku, Tokyo, Japan, **3** Educational System in Dentistry, Graduate School of Medical and Dental Sciences, Tokyo Medical and Dental University, Bunkyo-ku, Tokyo, Japan

* uto@ai.lab.uec.ac.jp

**Data Availability Statement:** The dataset and main programs underlying the results presented in the study are available from https://doi.org/10.5061/dryad.tmpg4f56q.

## Abstract

Objective structured clinical examinations (OSCEs) are a widely used performance assessment for medical and dental students. A common limitation of OSCEs is that the evaluation results depend on the characteristics of raters and a scoring rubric. To overcome this limitation, item response theory (IRT) models such as the many-facet Rasch model have been proposed to estimate examinee abilities while taking into account the characteristics of raters and evaluation items in a rubric. However, conventional IRT models have two impractical assumptions: constant rater severity across all evaluation items in a rubric and an equal interval rating scale among evaluation items, which can decrease model fitting and ability measurement accuracy. To resolve this problem, we propose a new IRT model that introduces two parameters: (1) a rater–item interaction parameter representing the rater severity for each evaluation item and (2) an item-specific step-difficulty parameter representing the difference in rating scales among evaluation items. We demonstrate the effectiveness of the proposed model by applying it to actual data collected from a medical interview test conducted at Tokyo Medical and Dental University as part of a post-clinical clerkship OSCE. The experimental results showed that the proposed model was well-fitted to our OSCE data and measured ability accurately. Furthermore, it provided abundant information on rater and item characteristics that conventional models cannot, helping us to better understand rater and item properties.

## Introduction

Objective structured clinical examinations (OSCEs) are widely used for evaluating the clinical skills, knowledge, and attitudes of medical and dental students in a standardized and objective manner [1–7]. OSCEs are classified as a performance assessment, in which expert raters assess examinee outcomes or the processes involved in performing tasks.

**Funding:** this work was supported by JSPS KAKENHI Grant Number 19H05663. The acronym JSPS stands for "Japan Society for the Promotion of Science." We also confirm that the funder had no role in study design, data collection and analysis, decision to publish, or preparation of the manuscript.

**Competing interests:** The authors have declared that no competing interests exist.

A common limitation of OSCEs is that the evaluation results depend on rater characteristics such as severity, consistency, and range restriction [1–14]. Rater severity is the systematic inclination to assign higher or lower scores compared with other raters, while rater consistency denotes the degree to which raters maintain their criteria across examinees. Range restriction refers to the limited variability in scores assigned by a rater. Various strategies, including rater training and the use of scoring rubrics, have been adopted to mitigate the effects of these rater characteristics. Nevertheless, residual rater effects can persist despite such efforts, impacting the accuracy of ability measurement [14].

Moreover, OSCEs often employ a scoring rubric comprising multiple evaluation items. In such cases, the assigned scores are affected by the characteristics of those evaluation items, including item difficulty and discrimination [15]. Item difficulty refers to the relative difficulty level in obtaining higher scores in a given item, while item discrimination is the capability of an evaluation item to differentiate between examinees with varying levels of ability. Rating scales quantify performance levels for the score categories in each evaluation item. Thus, accounting for item characteristics, in addition to rater characteristics, is crucial for the accurate measurement of examinees' abilities [15].

To overcome these inherent limitations, item response theory (IRT) models have been proposed to estimate examinee abilities while taking into account the effects of rater and item characteristics [10, 14, 16–19]. One representative model is the many-facet Rasch model (MFRM) [17], which has been applied to various performance assessments [4, 8, 11, 20–25], including OSCEs [1–7, 9], in order to investigate the characteristics of raters and items as well as estimate examinee ability while mitigating the influence of those characteristics. However, the MFRM makes some strong assumptions about rater and item characteristics [12, 16, 25–27]. For instance, it assumes that raters exhibit a uniform level of consistency and that items possess equivalent discriminatory power, conditions that are not commonly observed in clinical practice. Thus, several extensions of the MFRM have been recently proposed that relax those assumptions [14, 16, 18, 25]. A recent example is the generalized MFRM (GMFRM) [25], which jointly considers the abovementioned rater and item characteristics: rater severity, consistency, and range restriction, as well as item difficulty and discrimination. The GMFRM is expected to provide better model fitting and more accurate ability measurement compared with the MFRM in cases where a variety of rater and item characteristics is assumed to exist. However, the GMFRM retains the following assumptions, which may not be satisfied under OSCE assessments.

1. It assumes no interaction between raters and items, meaning that the rater severity is assumed to be consistent across all evaluation items.

2. It assumes an equal interval rating scale among evaluation items, meaning that the relative difficulty of transitioning between adjacent score categories is consistent for all items.

These model assumptions can decrease model fitting and ability measurement accuracy when applied to OSCE data that do not satisfy them. Furthermore, the inability to capture these characteristic differences might hinder our understanding of some inherent characteristics of raters and items, while a clearer understanding of them is crucial for improving the assessment design and the accuracy of ability measurements.

To resolve these limitations, we propose an extension of the GMFRM that introduces two types of parameters: (1) an interaction parameter between items and raters, which represents the rater severity for each evaluation item, and (2) an item-specific step-difficulty parameter, which captures the relative difficulty of transitioning between adjacent score categories in each evaluation item.

We demonstrate the effectiveness of the proposed model through simulation experiments and application to actual data. In the actual data experiments, we apply the proposed model to rating data collected from a medical interview test, which was conducted at Tokyo Medical and Dental University in 2021 as part of a post-clinical clerkship OSCE (Post-CC OSCE) for sixth-year dental students. Using actual OSCE data, we demonstrate that the proposed model improves model fitting and ability measurement accuracy compared with conventional models. Furthermore, we show that the proposed model provides abundant information on rater and item characteristics that conventional models cannot, helping us to better understand rater and item properties.

## Materials and methods

### Data

The OSCE data we used consist of rating scores assigned by five raters to 30 videos recorded as material for reconfirmation of medical interview tests for sixth-year dental students. The medical interview tests were conducted with the aim of evaluating the students' level of achievement after clinical training at Tokyo Medical and Dental University in 2021. The videos recorded at the medical interview tests for educational purpose were used secondarily for this study. The students were informed in writing of their right to refuse secondary use of the recorded videos for this study. The Dental Research Ethics Committee of Tokyo Medical and Dental University has stated in writing that this research falls outside the purview of the Ethical Guidelines for Medical Research Involving Human Subjects and is thus deemed 'not applicable.' This decision is documented in the minutes of the 2019 9th Ethics Review Committee meeting. The five raters were dentists with experience in OSCE evaluation. Of the 5 raters, 2 each evaluated all 30 examinees, while the remaining 3 each evaluated 10 different examinees. The raters performed their evaluations independently, without communicating or consulting with one another. The 30 evaluation items on the scoring rubric, shown in Table 1, were scored on a 4-point scale (4: excellent; 3: good; 2: acceptable; 1: unsatisfactory).

The rating data $U$ obtained from the above OSCE is formulated as a set of scores $x_{ijr}$, assigned by rater $r \in \mathcal{R} = \{1, \ldots, R\}$ to examinee $j \in \mathcal{J} = \{1, \cdots, J\}$ on evaluation items $i \in \mathcal{I} = \{1, \ldots, I\}$, as

$$U = \{x_{ijr} \in \mathcal{K} \cup \{-1\} \mid i \in \mathcal{I}, j \in \mathcal{J}, r \in \mathcal{R}\}, \quad (1)$$

where $\mathcal{K} = \{1, \ldots, K\}$ is the score categories, and $x_{ijr} = -1$ indicates missing data. In our OSCE data, $R = 5$, $J = 30$, $I = 30$, and $K = 4$.

In this study, IRT is applied to the above data in order to investigate the characteristics of raters and evaluation items, and to accurately estimate the ability of examinees while taking into account the effects of rater and item characteristics. Note that the IRT model we propose is applicable not only to the above-explained OSCE data but also to various rubric-based performance assessments whose data has the structure given in Eq (1).

### Rater and item characteristics

Performance assessments such as OSCEs are susceptible to rater effects, which impact the evaluation results, and these effects are influenced by the individual rater characteristics. Common rater characteristics are severity, consistency, and range restriction [1–7, 9, 12–14, 24, 28].

- *Severity* refers to the tendency of some raters to systematically assign higher or lower scores compared with other raters, irrespective of the examinee's actual performance.

**Table 1. Evaluation items in the scoring rubric.**

|    | Evaluation Items |
|----|------------------|
| 1  | Greeting |
| 2  | Self-introduction |
| 3  | Patient identification |
| 4  | Medical interview explanation and consent |
| 5  | Confirmation of chief complaint |
| 6  | Confirmation of affected site(s) |
| 7  | Confirmation of when the complaint started |
| 8  | Confirmation of current symptoms (spontaneous pain, swelling, redness, agitation, purulent discharge) |
| 9  | Confirmation of evoked pain (cold pain, warm pain, occlusion pain) |
| 10 | Confirmation of the nature and severity of the evoked pain |
| 11 | Confirmation of history of hospital visits |
| 12 | Confirmation of whether there is any medication for the chief complaint |
| 13 | Confirmation of consultation behavior |
| 14 | Confirmation of symptom history up to the present day |
| 15 | Confirmation of history of anesthesia and tooth extraction and any abnormalities during those procedures |
| 16 | Confirmation of whether any systemic disease is present |
| 17 | Confirmation of necessary information for each disease (whether potential risks for dental treatment could be identified) |
| 18 | Confirmation of allergies (food, drugs) |
| 19 | Summarization and confirmation of the chief complaint |
| 20 | Confirmation of whether there is anything the examinee has forgotten to say |
| 21 | Cleanliness during examination |
| 22 | Efforts to understand the patient's situation (whether the examinee has confirmed the patient's wishes, status of hospital visits, financial situation, etc.) |
| 23 | Smooth flow of conversation |
| 24 | Appropriateness of the main points of the conversation (whether the judgment on the main points of the conversation was appropriate according to the patient's situation) |
| 25 | Eye contact, pauses, listening |
| 26 | Appropriateness of open questions |
| 27 | Respectful language, attention to terminology |
| 28 | Interaction with the patient |
| 29 | Listening to the patient's concerns |
| 30 | Overall rating |

- *Consistency* is the degree to which raters maintain their scoring criteria over time and across examinees. Consistent raters exhibit stable scoring patterns, making their evaluations more reliable and predictable. In contrast, inconsistent raters display varying scoring tendencies.

- *Range restriction* refers to the limited variability in scores assigned by a rater, often due to their reluctance to use the full range of the rating scale. This can result in a compression or clustering of scores, making it difficult to distinguish between examinees with varying performance levels.

Differences in these rater characteristics may arise due to differences in raters' expectations, training, interpretation of assessment criteria, or personal biases and can lead to an inaccurate estimation of examinees' abilities [14, 25, 29].

Furthermore, when we use a scoring rubric comprising multiple evaluation items, the assigned scores also depend on the characteristics of those evaluation items, including item difficulty, discrimination, and rating scale [11, 15].

- *Difficulty* refers to the relative ease or challenge presented by a given evaluation item, which affects the likelihood of examinees obtaining high scores. Items with high difficulty levels tend to have lower scores, whereas easier items tend to have higher scores.

- *Discrimination* is the ability of an evaluation item to differentiate between examinees with varying levels of ability. Items with high discrimination can effectively distinguish between high- and low-performing examinees, providing valuable information about their relative abilities. Items with low discrimination may not be as useful in differentiating examinee performance because they are more likely to give random scores irrespective of examinee ability.

- *Rating scale* quantifies performance levels for the score categories in each evaluation item. Rating scales may vary depending on the item. For example, it may be possible to find evaluation items with overused mid-point score categories, while others use the full range of score categories evenly.

It is important to accurately measure ability while taking into account these item and rater characteristics. Therefore, this study explores the application of IRT to estimate such characteristics and assess examinee ability while taking into account the effects of these characteristics.

## Item response theory and many-facet Rasch models

IRT [30] has recently been used for scoring and analysis in various assessments, especially high-stakes and large-scale testing scenarios. IRT uses probabilistic models called IRT models, which give the probability of an examinee's responses to a test item as a function of the examinee's latent ability and the item's characteristic parameters. The Rasch model and the two-parameter logistic model are the most widely used IRT models, and they are applicable to test items for which responses are scored in binary terms as correct or incorrect. Furthermore, there are various polytomous IRT models that are applicable to ordered categorical score data, including the rating-scale model [31], the partial-credit model [32], and the generalized partial-credit model [33]. These traditional IRT models are applicable to two-way data consisting of *examinees × test items*, and offer the following benefits:

1. Examinee ability can be estimated while taking into account test item characteristics, including difficulty and discrimination.

2. Examinee responses to different test items can be assessed on the same scale.

3. Item characteristics can be isolated from the examinees' characteristics, which helps in analyzing and maintaining the properties of a test and items.

However, these traditional models cannot be applied directly to OSCE data, in which the examinees' responses are scored by multiple raters on multiple items, even if we regard the parameters of test items as parameters of evaluation items in a rubric. This is because the assumed OSCE data are defined as three-way data consisting of *examinees × items × raters*.

Extended IRT models for such multi-faceted data have been proposed to address this problem [14, 16–19, 28]. The most common IRT model applicable to such data is the MFRM [17],

which defines the probability that rater $r$ assigns score $k$ to examinee $j$ on item $i$ as

$$P_{ijrk} = \frac{\exp \sum_{m=1}^{k}[D(\theta_j - \beta_i - \beta_r - d_m)]}{\sum_{l=1}^{K} \exp \sum_{m=1}^{l}[D(\theta_j - \beta_i - \beta_r - d_m)]}, \qquad (2)$$

where $\theta_j$ represents the latent ability of examinee $j$, $\beta_i$ represents the difficulty of item $i$, $\beta_r$ represents the severity of rater $r$, $d_m$ represents the difficulty of transition between scores $m - 1$ and $m$, and $K$ indicates the number of categories. $D = 1.7$ is the scaling constant used to minimize the difference between the normal and logistic distribution functions. For model identification, $\sum_{r=1}^{R} \beta_r = 0$, $d_1 = 0$, $\sum_{m=2}^{K} d_m = 0$, and a normal prior for the ability $\theta_j$ are assumed.

However, the MFRM relies on the following strong assumptions, which are rarely satisfied in practice [8, 12, 16, 25–27].

1. All items have the same discriminatory power.

2. All raters have the same assessment consistency.

3. There is no difference in range restriction among the raters.

Accordingly, various extensions of the models have been proposed that relax these assumptions [18, 19, 26, 27, 34, 35]. One recent extension model that relaxes all three assumptions simultaneously is the generalized MFRM (GMFRM) [25]. In the GMFRM, the probability that rater $r$ assigns score $k$ to examinee $j$ on item $i$ is defined as

$$P_{ijrk} = \frac{\exp \sum_{m=1}^{k}[D\alpha_i\alpha_r(\theta_j - \beta_i - \beta_r - d_{rm})]}{\sum_{l=1}^{K} \exp \sum_{m=1}^{l}[D\alpha_i\alpha_r(\theta_j - \beta_i - \beta_r - d_{rm})]}, \qquad (3)$$

where $\alpha_i$ represents the discriminatory power of item $i$, $\alpha_r$ represents the consistency of rater $r$, and $d_{rm}$ is the rater-specific step-difficulty parameter denoting the severity of rater $r$ of transition from score $m - 1$ to $m$, making it possible to reflect the range restriction for each rater. For model identification, $\prod_{i=1}^{I} \alpha_i = 1$, $\sum_{i=1}^{I} \beta_i = 0$, $d_{r1} = 0$, $\sum_{m=2}^{K} d_{rm} = 0$, and a normal prior for the ability $\theta_j$ are assumed.

The GMFRM can represent various rater and item characteristics, so it is expected to provide better model fitting and more accurate ability measurement compared with the conventional MFRM, especially when various rater and item characteristics are assumed to exist [25]. However, the GMFRM retains the following assumptions, which might not be satisfied in a practical OSCE-based evaluation.

1. It assumes no interaction between raters and items, meaning that the rater severity is assumed to be consistent across all items.

2. It assumes a rating scale with an equal interval among items, meaning that the relative difficulty of transitioning between adjacent score categories is consistent for all items.

Violating these assumptions would decrease model fitting and ability measurement accuracy when the model is applied to OSCE data that do not satisfy them. Furthermore, we might fail to interpret some inherent characteristics of raters and items that would be important for understanding and improving the accuracy of ability measurement.

## Proposed IRT model

To resolve the above limitations, we propose an extension of the GMFRM that introduces two parameters: (1) a rater–item interaction parameter representing the rater severity for each

evaluation item and (2) an item-specific step-difficulty parameter representing the difference in rating scales among evaluation items. The proposed model defines the probability $P_{ijrk}$ as

$$P_{ijrk} = \frac{\exp \sum_{m=1}^{k}[D\alpha_i \alpha_r (\theta_j - \beta_{ir} - d_{im} - d_{rm})]}{\sum_{l=1}^{K} \exp \sum_{m=1}^{l}[D\alpha_i \alpha_r (\theta_j - \beta_{ir} - d_{im} - d_{rm})]}, \tag{4}$$

where $\beta_{ir}$ is the rater–item interaction parameter that indicates the severity of rater $r$ on item $i$. Moreover, $d_{im}$ is the item-specific step-difficulty parameter that represents the difficulty of transition between scores $m - 1$ and $m$ for item $i$. Here, for model identification, we assume $\sum_{r=1}^{R} \log\alpha_r = 0$, $d_{r1} = 0$, $\sum_{m=2}^{K} d_{rm} = 0$, $d_{i1} = 0$, $\sum_{m=2}^{K} d_{im} = 0$, and a normal prior distribution for the ability $\theta_j$.

A unique feature of the proposed model is the addition of the item–rater interaction parameter $\beta_{ir}$, which makes it possible to capture the difference in rater severity among items. Another feature is the incorporation of an item-specific step-difficulty parameter $d_{im}$, enabling the model to represent the differences in rating scales among items. These improvements are expected to offer the following benefits.

1. Model fitting to the data and ability measurement accuracy are expected to be improved when an interaction between items and raters exists and rating scales differ depending on the item.

2. It is possible to analyze rater and item characteristics, which cannot be analyzed using MFRM and GMFRM, thereby helping us better understand rater and item properties.

**Model identifiability.**   As described above, the proposed model assumes $\sum_{r=1}^{R} \log\alpha_r = 0$, $d_{r1} = 0$, $\sum_{m=2}^{K} d_{rm} = 0$, $d_{i1} = 0$, $\sum_{m=2}^{K} d_{im} = 0$, and a normal prior distribution for the ability $\theta_j$ for model identification. To explain why these constraints are necessary, we transform the proposed model as follows.

$$\begin{aligned} P_{ijrk} &= \frac{\exp \sum_{m=1}^{k}[D\alpha_i \alpha_r (\theta_j - \beta_{ir} - d_{im} - d_{rm})]}{\sum_{l=1}^{K} \exp \sum_{m=1}^{l}[D\alpha_i \alpha_r (\theta_j - \beta_{ir} - d_{im} - d_{rm})]} \\ &= \frac{\exp\left[kD\alpha_i \alpha_r \theta_j - kD\alpha_i \alpha_r \beta_{ir} - D\alpha_i \alpha_r \sum_{m=1}^{k}(d_{im} + d_{rm})\right]}{\sum_{l=1}^{K} \exp\left[kD\alpha_i \alpha_r \theta_j - kD\alpha_i \alpha_r \beta_{ir} - D\alpha_i \alpha_r \sum_{m=1}^{k}(d_{im} + d_{rm})\right]} \end{aligned}$$

From the first form of the equation, we can confirm that a location indeterminacy exists among the four parameters $\theta_j$, $\beta_{ir}$, $d_{im}$, and $d_{rm}$. We deal with this indeterminacy by applying zero-sum constraints for $d_{im}$ and $d_{rm}$ and giving a normal distribution with a fixed mean for $\theta_j$.

From the second form of the above equation, we can confirm that a scale indeterminacy exists within each term. For the first term, $kD\alpha_i \alpha_r \theta_j$, we resolve the scale indeterminacy by fixing $\prod_{r=1}^{R} \alpha_r$ by $\sum_{r=1}^{R} \log\alpha_r = 0$ and giving a normal distribution with a fixed variance for $\theta_j$. Consequently, the scales of $\alpha_i$ and $\alpha_r$ are identified. Therefore, the scale of $\beta_{ir}$ in the second term $kD\alpha_i \alpha_r \beta_{ir}$ and $\sum_{m=1}^{k}(d_{im} + d_{rm})$ in the third term $D\alpha_i \alpha_r \sum_{m=1}^{k}(d_{im} + d_{rm})$ are also identifiable.

**Parameter estimation.**   We use an expected a posteriori (EAP) estimation, a type of Bayesian estimation based on a Markov chain Monte Carlo (MCMC) algorithm, for the parameter estimation of the proposed model because it is known to provide more robust results for complex models [15, 25, 36–40] compared with marginal maximum likelihood estimation using an expectation–maximization algorithm or maximum a posteriori estimation using a Newton–

Raphson algorithm, both of which are traditional IRT parameter-estimation methods [41]. Although the Metropolis-Hastings-within-Gibbs sampling method [35] is a commonly used MCMC algorithm for IRT models, the No-U-Turn (NUT) sampler [42], which is a variant of the Hamiltonian Monte Carlo (HMC) algorithm, has recently become a popular and efficient alternative MCMC algorithm [15, 25, 43, 44]. Therefore, we use a NUT-based MCMC algorithm. The estimation program was implemented using the software package RStan [45]. The EAP estimates are calculated as the mean of the parameter samples obtained from 2,000 to 5,000 periods, using three independent chains. We set the prior distributions for the proposed model as follows.

$$\begin{cases} \theta_j, \beta_{ir}, d_{im}, d_{rm} \sim N(0,1) \\ \alpha_r, \alpha_i \sim LN(0,0.5) \end{cases} \tag{5}$$

## Results

We demonstrate the effectiveness of the proposed model through simulation experiments and application to actual data.

### Parameter-recovery experiments

This subsection describes a simulation experiment for the parameter-recovery evaluation, which was conducted to confirm whether the proposed model parameters can be estimated appropriately based on the NUT-based MCMC algorithm. To this end, the following experiment was performed for different numbers of examinees $J \in \{30, 50\}$, items $I \in \{10, 30\}$, and raters $R \in \{5, 10\}$, with the different numbers determined based on the sizes of the actual OSCE data.

1. For $J$ examinees, $I$ items, and $R$ raters, randomly generate true model parameters from the distributions given in Eq (5). The number of score categories $K$ was fixed at 4 to match the condition of the actual data.

2. Given the true parameters, randomly generate rating data from the proposed model.

3. Estimate the model parameters from the generated data with the NUT-based MCMC algorithm.

4. Calculate the root mean square errors (RMSEs) and the biases between the estimated and true parameters.

5. Repeat the above procedure 50 times, and calculate the mean values of the RMSEs and biases.

Table 2 shows the results, which clearly indicate that the RMSEs decrease with increasing amounts of data per parameter. Specifically, we can confirm the following tendencies.

1. The RMSEs for the examinee ability $\theta_j$ tend to decrease as the number of items or raters increases.

2. The RMSEs for the item–rater interaction parameters $\beta_{ir}$ tend to decrease as the number of examinees increases.

3. The RMSEs for the item-specific parameters tend to decrease as the number of examinees or raters increases.

**Table 2. Results of parameter-recovery experiments.**

| J | I | R | RMSE | | | | | | Bias | | | | | |
|---|---|---|---|---|---|---|---|---|---|---|---|---|---|---|
| | | | $\theta_j$ | $\alpha_r$ | $\alpha_i$ | $\beta_{ir}$ | $d_{im}$ | $d_{rm}$ | $\theta_j$ | $\alpha_r$ | $\alpha_i$ | $\beta_{ir}$ | $d_{im}$ | $d_{rm}$ |
| 30 | 10 | 5 | .175 | .147 | .233 | .295 | .316 | .267 | .002 | .032 | .038 | -.007 | .000 | .000 |
| | | 10 | .141 | .146 | .191 | .298 | .173 | .165 | -.023 | .021 | .018 | -.022 | .000 | .000 |
| | 30 | 5 | .111 | .089 | .251 | .290 | .488 | .466 | -.016 | .016 | .079 | -.017 | .000 | .000 |
| | | 10 | .081 | .084 | .158 | .282 | .333 | .312 | .010 | .012 | .024 | .007 | .000 | .000 |
| 50 | 10 | 5 | .172 | .090 | .188 | .255 | .262 | .225 | -.015 | .013 | .017 | -.013 | .000 | .000 |
| | | 10 | .127 | .115 | .121 | .236 | .139 | .145 | .016 | .015 | -.010 | .015 | .000 | .000 |
| | 30 | 5 | .112 | .059 | .189 | .241 | .465 | .457 | .006 | .008 | .062 | .007 | .000 | .000 |
| | | 10 | .085 | .068 | .133 | .235 | .341 | .329 | .007 | .011 | .030 | .010 | .000 | .000 |

A result of .000 indicates that the absolute value was less than .001.

4. The RMSEs for the rater-specific parameters tend to decrease as the number of examinees or items increases.

Related studies [25, 29, 46] have also shown that the RMSEs decrease with increasing amounts of data per parameter, which is consistent with our results. Furthermore, as shown in the table, the RMSEs for examinee ability $\theta_j$ are less than 0.2. Considering that standard normal distribution is assumed for $\theta_j$, an RMSE value below 0.2 is generally acceptable because it corresponds to only about 3% of the logit range [-3,3], where 99.7% of $\theta_j$ falls statistically.

Table 2 also shows that the average bias was nearly zero overall, indicating that there was no overestimation or underestimation of the parameters. Furthermore, we confirmed the Gelman–Rubin statistic $\hat{R}$ [47, 48], a well-known convergence diagnostic index, and the effective sample size (ESS). The $\hat{R}$ values were less than 1.1 in all cases, indicating that the MCMC runs converged. Moreover, an ESS of more than 400 is considered sufficiently large [49], and our ESSs satisfied this criterion in all MCMC runs.

Based on these results, we conclude that the parameter estimation for the proposed model can be appropriately performed using the MCMC algorithm.

## Actual data experiments

In this section, we evaluate the effectiveness of the proposed model through experiments using the actual data that we collected from a medical interview test conducted as part of a Post-CC OSCE at Tokyo Medical and Dental University, as detailed in *Data* Section.

**Model comparison using information criteria.** We first conducted model-comparison experiments using information criteria. As the information criteria, we used the widely applicable information criterion (WAIC) [50] and the widely applicable Bayesian information criterion (WBIC) [51], which are applicable for Bayesian estimation using MCMC.

We compared the information criteria for the proposed model with those for the conventional models, namely, GMFRM and MFRM. Furthermore, to directly evaluate the effectiveness of the newly incorporated parameters $\beta_{ir}$ and $d_{im}$, we examined the following two restricted versions of the proposed model.

- A proposed model that decomposes the item–rater interaction parameter $\beta_{ir}$ into $\beta_i + \beta_r$.

- A proposed model without item-specific step-difficulty parameters $d_{im}$.

**Table 3. Model comparison based on information criteria and ability measurement accuracy.**

| | WAIC | WBIC | Ability measurement accuracy | | | | | |
| --- | --- | --- | --- | --- | --- | --- | --- | --- |
| | | | $N = 20$ | | | $N = 40$ | | |
| | | | Avg. | SD | *p*-value | Avg. | SD | *p*-value |
| Proposed model | **3346.0** | 2272.4 | **.893** | .028 | - | **.956** | .016 | - |
| • with $\beta_{ir} = \beta_i + \beta_r$ | 3408.6 | **2101.2** | .871 | .033 | <.01 | .945 | .018 | <.01 |
| • w/o $d_{im}$ | 3783.6 | 2427.5 | .874 | .032 | <.01 | .947 | .018 | <.01 |
| GMFRM | 383.9 | 2239.2 | .842 | .042 | <.01 | .930 | .021 | <.01 |
| MFRM | 4423.4 | 2389.7 | .829 | .042 | <.01 | .922 | .025 | <.01 |
| Score averaging method | - | - | .809 | .046 | <.01 | .914 | .023 | <.01 |

Table 3 shows the WAIC and WBIC values calculated for these models, where the model that minimizes the criteria is regarded as optimal. In the table, bold font indicates the model that minimizes each criterion. According to the results, WAIC selected the proposed model as the best model, whereas WBIC selected the proposed model that decomposes $\beta_{ir} = \beta_i + \beta_r$ as the best model. Although WBIC did not select the proposed model as the best model, a comparison of the proposed model and that without $d_{im}$ shows that removing $d_{im}$ substantially deteriorates the WBIC. These results suggest that the item-specific step-parameter $d_{im}$ is essential for improving model fitting, although the effects of the item–rater interaction parameter $\beta_{ir}$ might be relatively small.

Additionally, we checked the proposed model's goodness of fit to the data. For this, we used a posterior predictive *p*-value (*PPP*-value) [47]. Specifically, we calculated a *PPP*-value for the proposed model by using an averaged standardized residual (a traditional metric of IRT model fitness under a non-Bayesian framework) as a discrepancy function, in a manner similar to that in [52–54]. The *PPP*-value is around 0.5 for a well-fitted model but can be extremely low or high, taking values less than 0.05 or greater than 0.95, for a poorly fitted model. The *PPP*-value of the proposed model was 0.55, which is near 0.5, suggesting that the model is well-fitted to the data.

**Comparison of ability measurement accuracy.** This subsection compares the accuracy of ability measurement, using the abovementioned actual data. Specifically, we evaluate the extent to which ability estimates are stable when abilities are estimated using data from different items and raters. If a model appropriately reflects item and rater characteristics, ability values estimated from data comprising different items and raters should be highly stable. This concept is based on the split-half method [55], a well-known method for measuring the accuracy of ability measurement in test theory. Specifically, we conducted the following experiment for the proposed model and the comparative models, which were examined in the above model-comparison experiment.

1. Estimate the model parameters from the full dataset, using MCMC.

2. Randomly select $N = 20$ or 40 scores assigned to each examinee, then change the unselected scores to missing data.

3. Estimate examinee abilities using the dataset with missing data, and the rater and item parameters estimated in Procedure 1.

4. Calculate the correlation between the ability estimates obtained in Procedure 3 and those obtained in Procedure 1.

5. Repeat Procedures 2 to 4 50 times and then calculate the average and standard deviation of the correlations.

To determine a baseline with a non-IRT approach, we conducted the same experiment, using a method in which the ability estimates are given as simple average scores. We designated this as *the score averaging method*. We also conducted multiple comparisons using Dunnett's test to ascertain whether correlation values under the proposed model are significantly higher than those under the other models and the score averaging method.

Table 3 shows the results, which indicate that all IRT models provide higher average correlation values compared with the score averaging method, suggesting that the IRT models effectively improve the accuracy of ability measurements. The results also show that the proposed model provides significantly higher correlations compared with the other models, demonstrating the proposed model's superior accuracy in estimating ability.

**Parameter interpretation.** Besides the improvement in model fitting and ability measurement accuracy, another benefit of the proposed model is its higher interpretability on rater and item characteristics compared with conventional models. This subsection interprets those characteristics based on the proposed model parameters estimated from the actual data.

First, we show the histogram of the ability estimates in Fig 1. The horizontal axis indicates the ability values $\theta$, the vertical axis indicates the probability density values, the bars show the histogram, and the solid blue line shows the probability density function estimate. From the results, we can confirm that the ability estimates are scattered around 0, where the minimum

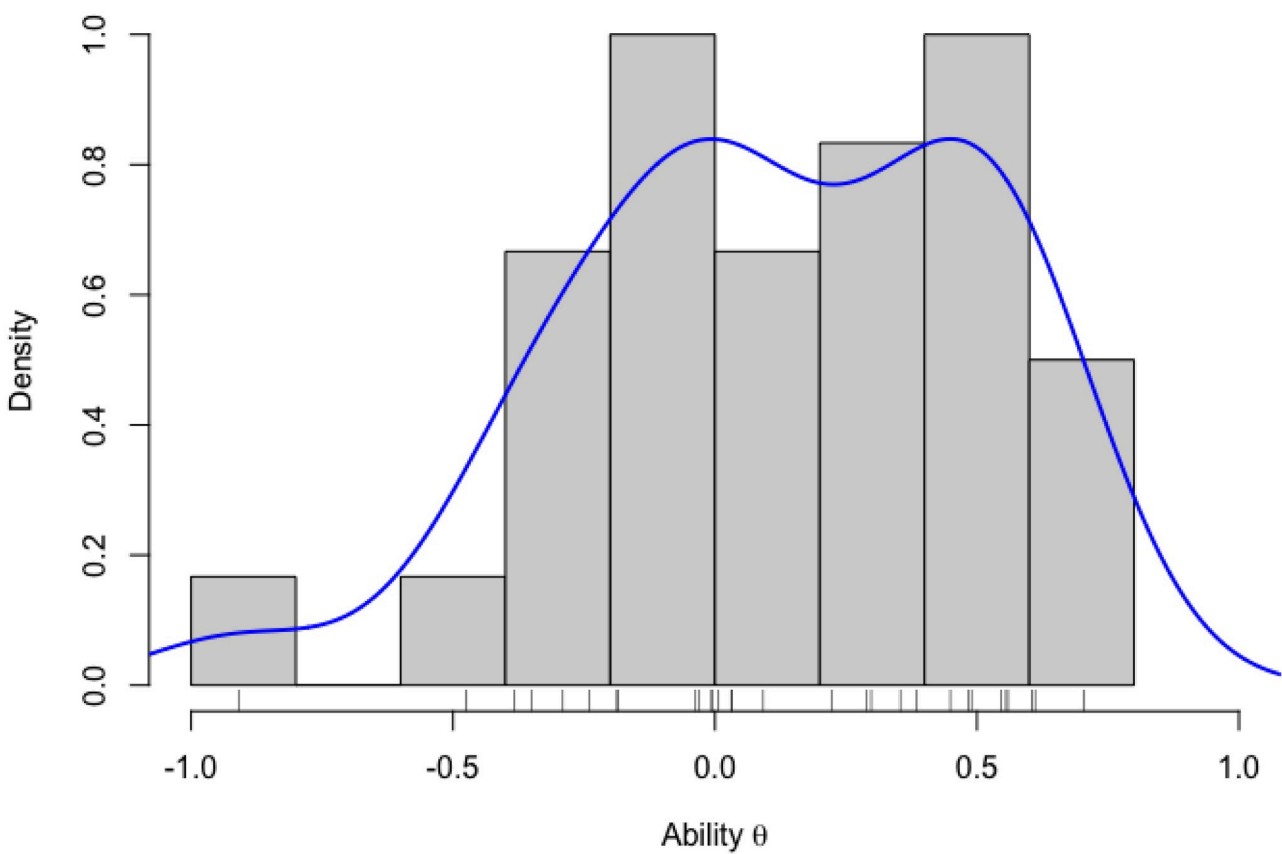

**Fig 1. Histogram of ability estimates.**

**Table 4. Estimates of rater parameters.**

| $r$ | $\alpha_r$ | $d_{r2}$ | $d_{r3}$ | $d_{r4}$ | $Avg(\boldsymbol{\beta}_r)$ |
|---|---|---|---|---|---|
| 1 | 0.96 | -0.46 | -1.64 | 2.10 | 0.13 |
| 2 | 0.86 | -0.97 | -1.31 | 2.28 | 0.08 |
| 3 | 1.28 | -0.70 | -0.33 | 1.03 | -0.17 |
| 4 | 1.05 | -0.72 | -0.71 | 1.43 | -0.29 |
| 5 | 0.90 | -0.81 | -1.11 | 1.92 | 0.11 |

of the ability estimates was -0.91, the maximum was 0.70, the average was 0.12, and the standard deviation was 0.39.

The rater-parameter estimates are shown in Table 4, where $Avg(\boldsymbol{\beta}_r)$ indicates the overall severity of each rater calculated based on the item–rater interaction parameter $\beta_{ir}$ as

$$Avg(\boldsymbol{\beta}_r) = \frac{1}{I}\sum_{i=1}^{I} \beta_{ir}. \tag{6}$$

Furthermore, Fig 2 depicts the item response curves (IRCs) of the proposed model, which are plots of $P_{ijrk}$, for each rater, where the parameter of item 1 was given. In the IRCs, the horizontal axis shows the examinee ability $\theta_j$, the first vertical axis shows the response probability for each category, and the black lines with markers indicate the IRCs. The gray shaded area indicates the range in which the ability estimates exist, namely, $\theta \in [-0.91, 0.70]$. Furthermore, in the figure, the red solid line shows the standard error (SE) of ability, which indicates how accurately a given item and a given rater measure an examinee with an ability level, and the

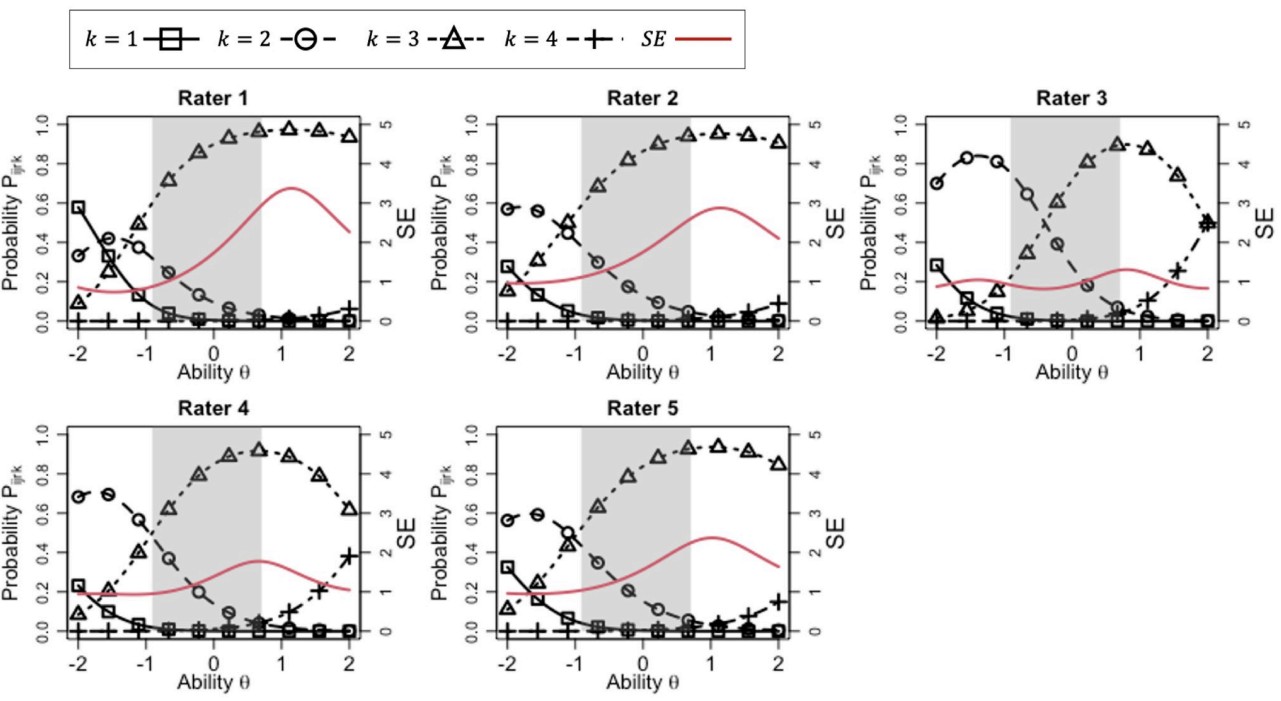

**Fig 2. IRCs for each rater in item 1.**

second vertical axis shows the standard error values. In IRT, the SE of ability is defined as the inverse square root of the Fisher information (FI), where more information implies less error in the assessment. In the proposed model, the FI of rater $r$ in item $i$ for an examinee $j$ with ability $\theta_j$ can be calculated as

$$I_{ijr} = D^2 \alpha_i^2 \alpha_r^2 \left[ \sum_{k=1}^{K} k^2 P_{ijrk} - \left( \sum_{k=1}^{K} k P_{ijrk} \right)^2 \right]. \tag{7}$$

Consequently, the SE can be calculated as $\sqrt{1/I_{ijr}}$. A smaller SE on a specific ability value $\theta$ indicates that the rater can distinguish the ability level more precisely.

From the IRCs, we can see that raters 1, 2, 4, and 5 tend to have a range restriction in which score category 3 is overused, while rater 3 tends to use two categories, 2 and 3, evenly. Moreover, the rater consistency parameters shown in Table 4 suggest that rater 3 tends to distinguish the examinees' ability levels more consistently. This characteristic can also be confirmed from the fact that the SE of rater 3 is relatively low in the ability range $\theta \in [-0.91, 0.70]$ (gray shaded area). Furthermore, the overall severity $Avg(\boldsymbol{\beta}_r)$ shows that rater 4 tends to be a little lenient, although the difference in overall severity is not large among the raters.

Next, we show the item-parameter estimates in the *item parameters* column in Table 5, where $Avg(\boldsymbol{\beta}_i)$ indicates the overall difficulty of each item calculated based on the item–rater interaction parameter $\beta_{ir}$ as

$$Avg(\boldsymbol{\beta}_i) = \frac{1}{R} \sum_{r=1}^{R} \beta_{ir}. \tag{8}$$

We also show that the IRCs and SE function for each item, given the parameters for rater 3, in Figs 3 and 4. According to the IRCs, it is clear that the rating scale differs depending on the item. For example, the following tendencies can be confirmed.

1. The highest score category tends to be overused strongly in items 3 and 5, while score category 3 is highly used in items 21 and 26. In those items, nearly all the target examinees had the same score, indicating that these items are ineffective for distinguishing the ability of examinees. This can also be confirmed from the toward relatively high SEs in the ability range shaded gray in the figures.

2. In items 2, 16, 19, 22, 23, 24, 25, 27, and 29, the IRCs for three score categories tend to show high probabilities in or near the ability range of the target examinees (the gray-shaded area), indicating that these items can be utilized to distinguish the ability of examinees according to these three score categories. Thus, these items tend to provide low SE values across a wide range of the target examinees' abilities, indicating their high effectiveness for the ability measurement.

3. The IRCs for the other items show relatively high probabilities for only two score categories within the ability range of the target examinees (the gray-shaded area), indicating that these items tend to distinguish the examinees' ability in two score categories. For example, item 23 distinguishes the examinees using scores 2 and 4, whereas items 13 and 14 do so using scores 1 and 3. Because of this range restriction, these items might not be adequate for evaluating the entire range of the target examinees' abilities, although they might help to distinguish a specific ability for which the IRCs of the two most-used score categories intersect.

**Table 5. Estimates of item parameters, and item–rater interaction parameters $\beta_{ir}$.**

| $i$ | Item parameters | | | | | Item–rater interaction parameter $\beta_{ir}$ | | | | |
|---|---|---|---|---|---|---|---|---|---|---|
| | $\alpha_i$ | $d_{i2}$ | $d_{i3}$ | $d_{i4}$ | $Avg(\beta_i)$ | $\beta_{i1}$ | $\beta_{i2}$ | $\beta_{i3}$ | $\beta_{i4}$ | $\beta_{i5}$ |
| 1 | 1.09 | -1.42 | 0.19 | 1.23 | -0.13 | 0.19 | -0.06 | -0.26 | -0.41 | -0.10 |
| 2 | 1.58 | -0.84 | -0.08 | 0.92 | 0.33 | 0.27 | 0.56 | 0.92 | -0.49 | 0.42 |
| 3 | 0.89 | 0.22 | 1.16 | -1.39 | -1.20 | -1.22 | -0.70 | -1.57 | -1.02 | -1.50 |
| 4 | 1.28 | 0.09 | -0.84 | 0.75 | 0.26 | 0.60 | 0.58 | 0.29 | -0.66 | 0.49 |
| 5 | 0.98 | 0.22 | 0.55 | -0.77 | -1.25 | -1.57 | -0.13 | -1.70 | -1.15 | -1.68 |
| 6 | 1.08 | -0.20 | -0.26 | 0.47 | -0.76 | -0.38 | -0.24 | -1.37 | -0.35 | -1.47 |
| 7 | 1.84 | 0.14 | -0.96 | 0.81 | -0.65 | -0.34 | -0.44 | -1.20 | -0.89 | -0.37 |
| 8 | 0.86 | -1.34 | 1.68 | -0.33 | 1.06 | 1.51 | 0.58 | 1.49 | 0.31 | 1.42 |
| 9 | 0.86 | 1.62 | 0.88 | -2.50 | -0.15 | -0.14 | -0.02 | -0.26 | -0.04 | -0.30 |
| 10 | 0.70 | -1.87 | 1.06 | 0.81 | 0.45 | 0.80 | 0.06 | 1.23 | -0.57 | 0.71 |
| 11 | 0.59 | 0.99 | -1.62 | 0.63 | 1.11 | 2.25 | 0.34 | 0.70 | 0.28 | 1.97 |
| 12 | 1.22 | 0.81 | -1.41 | 0.60 | 0.68 | 0.93 | 0.96 | 0.30 | 0.50 | 0.71 |
| 13 | 1.53 | 0.65 | -1.24 | 0.59 | 0.59 | 0.81 | 0.80 | 0.77 | -0.18 | 0.73 |
| 14 | 0.78 | 0.87 | -1.53 | 0.67 | 0.50 | 1.01 | 0.51 | 0.60 | -0.46 | 0.84 |
| 15 | 0.69 | 1.17 | 1.11 | -2.29 | -1.11 | -1.12 | -1.20 | -0.68 | -1.59 | -0.97 |
| 16 | 0.70 | 0.47 | -0.18 | -0.29 | -1.35 | -1.63 | -2.12 | -0.29 | -1.06 | -1.65 |
| 17 | 0.78 | 0.25 | -0.94 | 0.69 | 0.68 | 2.18 | -0.51 | -0.21 | -0.29 | 2.23 |
| 18 | 1.00 | 0.62 | -0.21 | -0.41 | 0.41 | 0.67 | 0.43 | 0.44 | -0.07 | 0.57 |
| 19 | 0.63 | -0.51 | -0.52 | 1.03 | 0.21 | 0.81 | 0.32 | -0.01 | -0.46 | 0.40 |
| 20 | 0.82 | 0.54 | -1.31 | 0.77 | 0.81 | 0.73 | 0.99 | 0.64 | 0.91 | 0.79 |
| 21 | 1.32 | -0.57 | -0.82 | 1.39 | -0.41 | -0.38 | -0.48 | -0.96 | -0.38 | 0.15 |
| 22 | 0.56 | 1.02 | -1.44 | 0.41 | 0.27 | 0.88 | -0.28 | -0.49 | 0.46 | 0.78 |
| 23 | 1.99 | -0.99 | 1.23 | -0.24 | -0.50 | -0.06 | -0.72 | -0.77 | -0.74 | -0.24 |
| 24 | 2.20 | -0.94 | 0.90 | 0.05 | -0.40 | -0.58 | 0.42 | -0.89 | -0.24 | -0.73 |
| 25 | 1.38 | -0.38 | 0.58 | -0.20 | -0.26 | -0.17 | 0.31 | -0.69 | -0.26 | -0.50 |
| 26 | 2.00 | -0.70 | -0.82 | 1.52 | -0.22 | -0.34 | 0.45 | -0.39 | -0.38 | -0.47 |
| 27 | 1.03 | -1.09 | -0.14 | 1.23 | -0.29 | -0.58 | 0.34 | -0.88 | -0.26 | -0.07 |
| 28 | 1.13 | -1.06 | 0.33 | 0.72 | -0.17 | -1.05 | 0.34 | -0.43 | 0.03 | 0.27 |
| 29 | 1.24 | -0.54 | -0.49 | 1.03 | 0.32 | -0.40 | 0.57 | 0.57 | 0.29 | 0.55 |
| 30 | 1.43 | -0.84 | -0.09 | 0.93 | 0.32 | 0.19 | 0.69 | 0.03 | 0.54 | 0.14 |

This can be confirmed from the fact that these items tended to show relatively low SE values around such ability points.

Table 5 shows that the item discrimination parameter $\alpha_i$ and the overall difficulty $Avg(\boldsymbol{\beta}_i)$ also vary depending on the item. For the ease of grasping the tendency, Fig 5 shows the item discrimination parameter $\alpha_i$ and the overall difficulty $Avg(\boldsymbol{\beta}_i)$ for each item, where the horizontal axis indicates the $\alpha_i$ values, the vertical axis indicates the $Avg(\boldsymbol{\beta}_i)$ values, and the plots indicate the item index. As shown in the figure, the $\alpha_i$ estimates suggest that items 11 and 22 have low discriminatory power, whereas items 23, 24, and 26 have extremely high discriminatory power. In addition, the overall difficulties suggest that items 3, 5, 15, and 16 are extremely easy, whereas items 8 and 11 are extremely difficult. We can also see that the variety of the item discriminatory powers and the overall item difficulty are substantially large compared with the rater consistency and overall severity among raters.

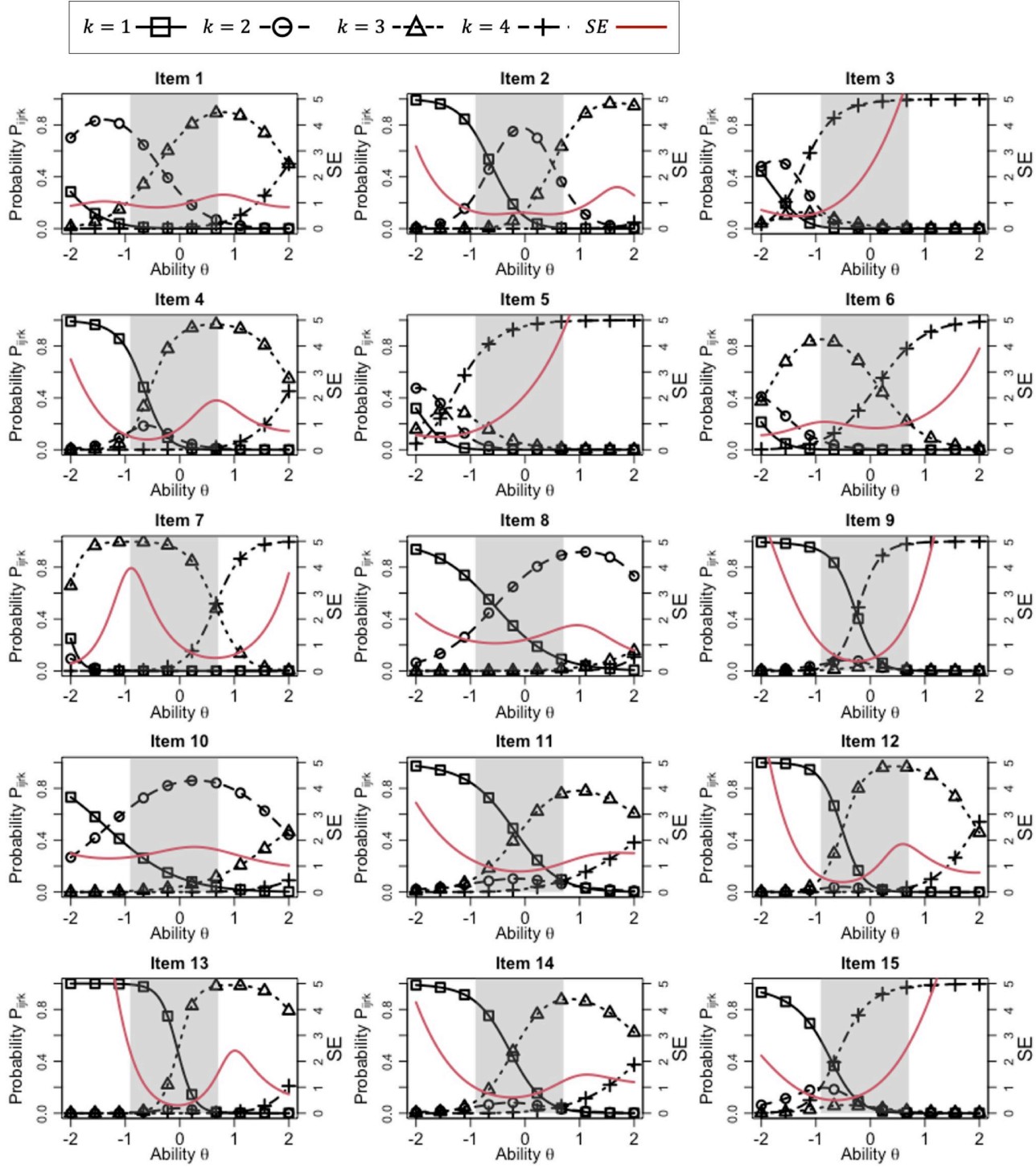

**Fig 3. IRCs for items 1–15, where rater parameters for *r* = 3 are given.**

Finally, we show the item–rater interaction parameters in the *item–rater interaction parameters* column in Table 5. We plot these values in Fig 6, where the horizontal axis indicates the item index, the vertical axis indicates the $\beta_{ir}$ values, each line indicates one rater, and the black x's indicate the overall item difficulty $Avg(\boldsymbol{\beta}_i)$. As the values and figure show, the severity of

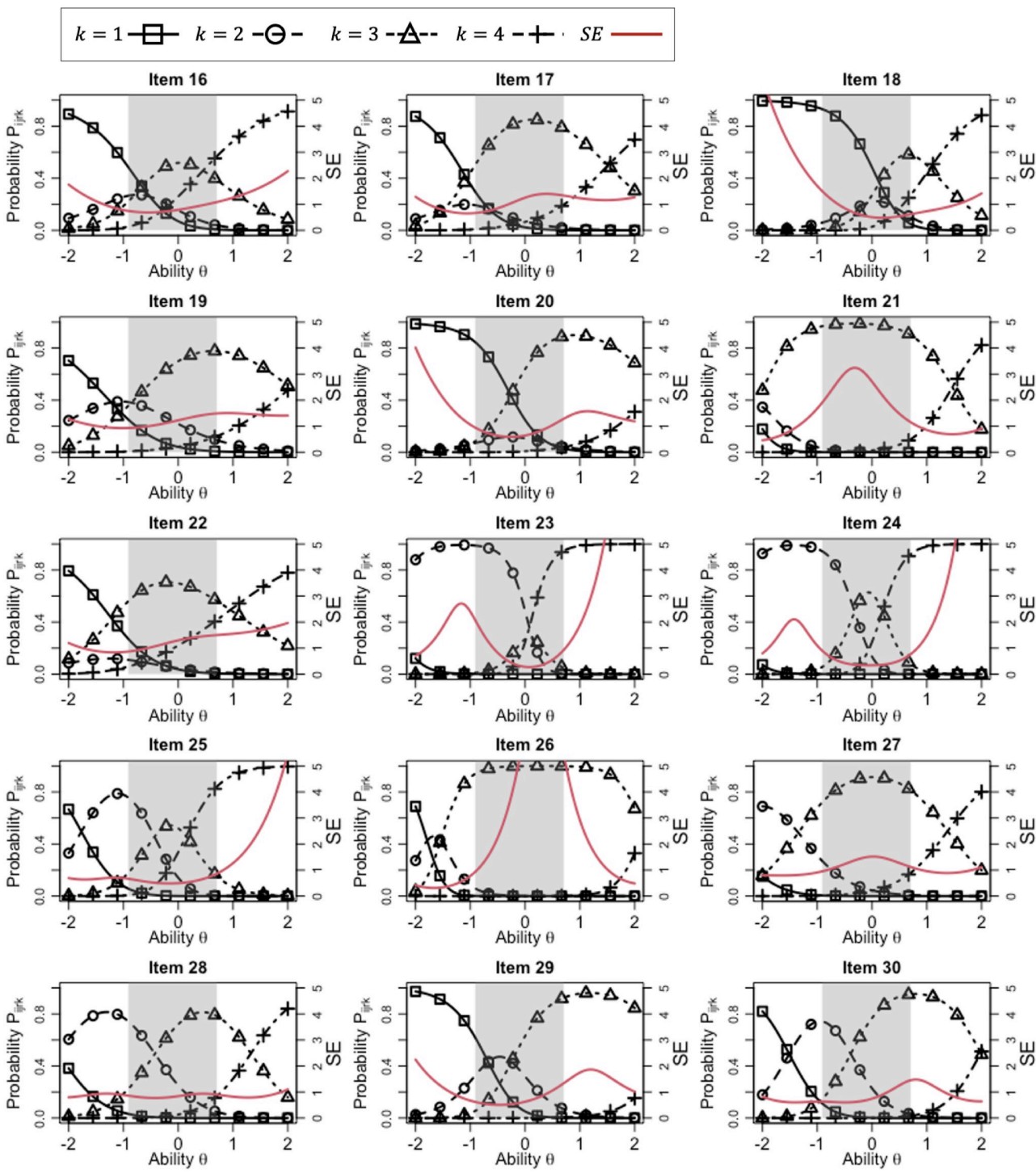

**Fig 4. IRCs for items 16–30, where rater parameters for $r = 3$ are given.**

each rater is inherently inconsistent across items. For example, rater 2 is more severe compared with the other raters on items 5, 24, 25, 26, and 27 but is more lenient on items 16, 17, and 22. Moreover, raters 1 and 5 seem to be extremely severe on items 11, 17, and 22 in particular.

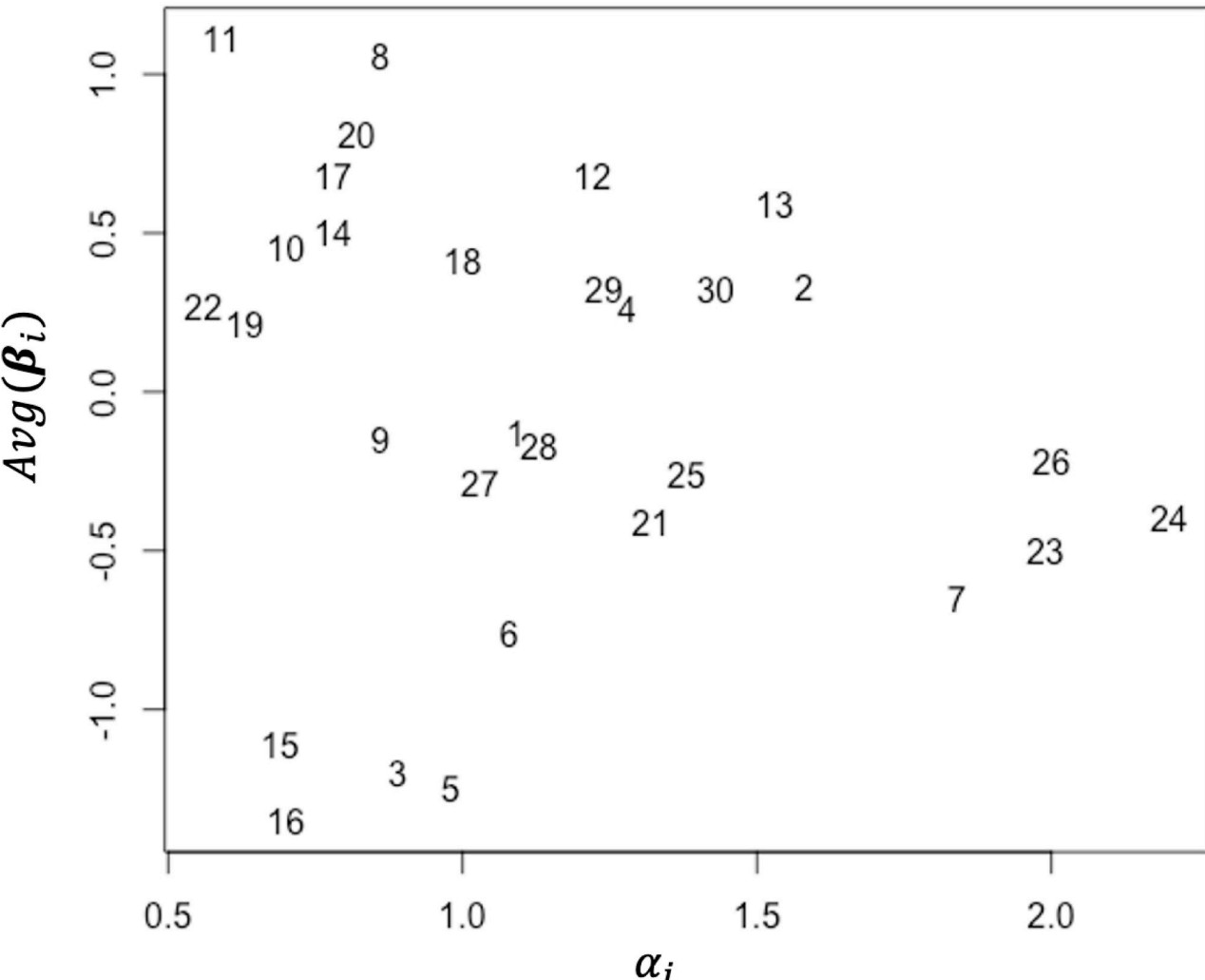

**Fig 5. Estimates of the discriminatory power and overall difficulty for each item.** The numbers in the graph indicate the item index.

## Discussion

The above experiments using actual data revealed that rater severities and rating scales varied among evaluation items. Unlike conventional models, the proposed model estimated examinee ability while considering the effects of these two characteristics, which explains why the proposed model showed improved model fitting and ability measurement accuracy.

### Implications of findings

Analysis of our OSCE data reveals the following observation about the rubric and raters. First, many evaluation items in the rubric cause a few restricted score categories to be overused, meaning that some score categories are not used efficiently. Specifically, items 3, 5, 21, and 26 induce the extreme overuse of one score category, thereby resulting in low SE values for the target ability range. These items might have to be revised to improve the effectiveness of the ability measurement and reduce the assessment burden of raters. Furthermore, the item–rater

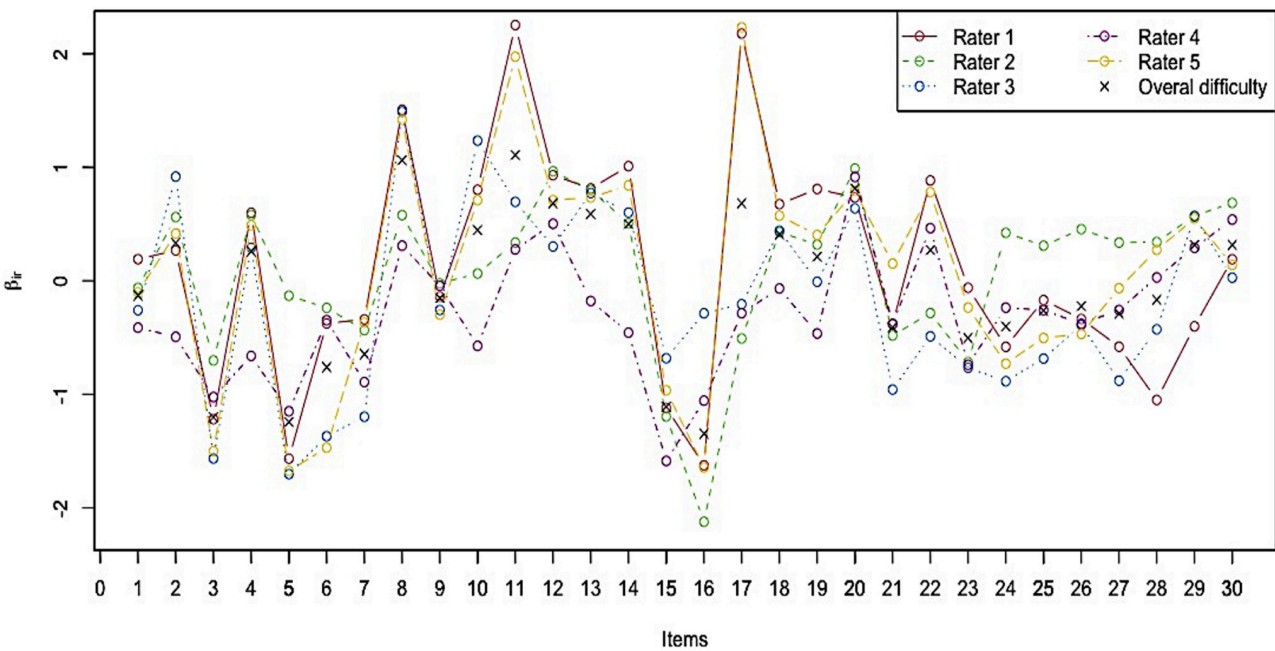

**Fig 6. Estimates of the item–rater interaction parameter $\beta_{ir}$.**

interaction parameters suggest that rater 2 tends to be substantially lenient only for evaluation items 24, 25, 26, and 27, which are related to communication skills. This may reflect the rater's unconscious scoring biases, and thus providing such feedback would be meaningful for improving their future rating behavior.

In addition, the proposed model excels at analyzing the consistency, overall severity, and range restriction of raters and the discrimination power and overall difficulty of items. Understanding such characteristics might also be useful for providing feedback to raters, designing rater training programs, and evaluating and revising a rubric's evaluation items, as detailed in the next paragraph. Checking the SE function is also beneficial for investigating the overall measurement accuracy for a specific ability level.

Although the direct contribution of our model to general clinical practice, other branches of clinimetrics, and routine patient care evaluations may be limited, our model has the following features, which are relevant to the field of clinimetrics and will ultimately lead to higher quality clinical practice and patient care:

- Appropriate Selection of Competent Professionals: Our model realizes accurate measurement of examinee abilities in the OSCE, even when multiple raters are involved. This contributes to the appropriate selection of competent medical and dental professionals, enhancing future patient care quality.

- Trust in Evaluation and Learning Processes: Accurate ability measurement enhances the trustworthiness of both the evaluation and the learning processes for medical and dental students. This trust is fundamental for the credibility and effectiveness of medical education.

- Improved Rubric Design and Development: Our model allows for a detailed analysis of the characteristics of each evaluation item in the rubric. This helps in refining and developing more effective rubrics, thereby ensuring fair and comprehensive assessment criteria.

- Enhanced Rater Training: Our model provides detailed and objective information on the characteristics of raters. Offering such information to each rater serves as valuable feedback for rater training programs. Such feedback can help raters, who are typically medical and dental professionals, become aware of their biases and standardize their evaluations, thereby improving the reliability of assessments.

- Reducing Rating Cost: By enabling high-quality rubric design and improving rater reliability, our model can potentially reduce the number of raters and stations while maintaining ability measurement accuracy, which contributes to cost reduction for managing examinations.

- Broader Applicability: Our model is applicable to various rubric-based performance assessments beyond OSCEs, including writing exams, interview exams, and presentation exams, in fields such as psychology, nursing, and allied health professions.

Note that the interpretation of item and rater characteristics based on the IRT model cannot be realized directly by basic statistics such as the mean or variance of scores for each item and rater given that such simple statistics cannot isolate the effects derived from the various characteristics of examinees, rater, and items.

## Limitations and future works

One limitation of this study is the relatively small size of the actual data, which might affect the parameter estimation accuracy. However, as shown in the parameter-recovery experiments, the proposed model provides acceptable parameter estimation accuracy given the size of the actual data, namely, $J = 30$, $I = 30$, and $R = 5$. Note that the MFRMs and its extension models, including the proposed model, use $I \times R = 150$ data points for estimating the examinee ability $\theta_j$. Similarly, the item and rater parameters are estimated from the $J \times R = 150$ and $J \times I = 900$ data points, respectively. This suggests that the amount of data for each parameter is not too small, although the effects of increasing data size should be investigated in future studies.

The proposed IRT model is applicable to various rubric-based performance assessments and offers various benefits as detailed above. However, to use our model appropriately, careful rater allocation is required for IRT parameter linking. The ideal condition for parameter linking is a setting where all examinees are assessed by all raters. However, in general settings, to reduce the scoring burden on each rater, each examinee is assessed by a few different raters from a pool of raters. To ensure parameter linking in such cases, each examinee must be assessed by at least two raters, and the combinations of raters must be changed several times throughout the examination. This is because when raters remain fixed, the factors of raters and examinees become nested, making it impossible to separate their characteristics. Under inappropriate rater allocation designs, fair ability measurement cannot be achieved by any statistical analysis method, including IRT, without some strong assumptions. For appropriate rater allocation designs, please refer to [28]. Note that our dataset follows an appropriate rater allocation design.

This study assumes the unidimensionality of ability, and the model–data fit based on *PPP* suggested that the unidimensional proposed model is well-fitted to the data. Meanwhile, a multidimensional extension might have further benefits. Thus, examining multidimensional variants of the proposed model will be another future task. It is important to note that the unidimensionality assumption and the relatively small sample size do not preclude the use of IRT in analyzing OSCE data. Although such conditions might lead to inaccurate or biased parameter estimation, the same difficulties arise when simple statistics such as mean scores are employed as ability estimates. Compared with non-IRT approaches, the proposed IRT model

has the advantage of being able to estimate examinee ability while considering various bias effects derived from raters and items, resulting in a higher ability measurement accuracy.

Another limitation of this study is that other rater characteristics, which may have affected the assessment results, were ignored. Some representative examples are rater-parameter drift [56–58], differential rater functioning [59–61], and the halo effect [12]. This study also ignored other possible bias factors sourced by stations and simulated/standardized patients [2, 4, 6, 7]. In future work, we will extend the proposed model to account for such complex factors.

## Conclusion

This study proposed a new IRT model that considers the characteristics of raters and rubric evaluation items. Specifically, the proposed model is formulated as a GMFRM extension with two additional parameters: the rater–item interaction parameter and the item-specific step-difficulty parameter. The proposed model relaxes two strong, unrealistic assumptions of conventional models, namely, the consistent rater severity across all evaluation items and the equal interval rating scale across all evaluation items. Through experiments using actual OSCE data, we showed that the proposed model succeeded in improving model fitting, ability measurement accuracy, and the interpretability of rater and item properties compared with conventional models.

## Author Contributions

**Conceptualization:** Masaki Uto, Kouji Araki, Maomi Ueno.

**Data curation:** Jun Tsuruta, Kouji Araki.

**Formal analysis:** Masaki Uto.

**Funding acquisition:** Maomi Ueno.

**Methodology:** Masaki Uto.

**Writing – original draft:** Masaki Uto.

**Writing – review & editing:** Masaki Uto, Jun Tsuruta, Kouji Araki, Maomi Ueno.

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
