## [Decision Letter · Decision Letter 0]

29 May 2024

PONE-D-24-02840Item response theory model highlighting rating scale of a rubric and rater-rubric interaction in objective structured clinical examinationPLOS ONE

Dear Dr. Uto,

Thank you for submitting your manuscript to PLOS ONE. After careful consideration, we feel that it has merit but does not fully meet PLOS ONE’s publication criteria as it currently stands. Therefore, we invite you to submit a revised version of the manuscript that addresses the points raised during the review process.

We look forward to receiving your revised manuscript.

Kind regards,

Leona Cilar Budler

Academic Editor

PLOS ONE

Journal Requirements:

 [This work was supported by JSPS KAKENHI Grant Number 19H05663.].  

5. Please expand the acronym “JSPS” (as indicated in your financial disclosure) so that it states the name of your funders in full.

6. Thank you for uploading your study's underlying data set. Unfortunately, the repository you have noted in your Data Availability statement does not qualify as an acceptable data repository according to PLOS's standards.

Additional Editor Comments:

Your paper was reviewed by two reviewers. They suggested some additional comments to improve paper quality. Overall, there are just minor issues which could be revised.

Reviewers' comments:

Reviewer's Responses to Questions

**Comments to the Author**

1. Is the manuscript technically sound, and do the data support the conclusions?

Reviewer #1: Yes

Reviewer #2: Yes

2. Has the statistical analysis been performed appropriately and rigorously? 

Reviewer #1: Yes

Reviewer #2: Yes

3. Have the authors made all data underlying the findings in their manuscript fully available?

Reviewer #1: Yes

Reviewer #2: Yes

4. Is the manuscript presented in an intelligible fashion and written in standard English?

Reviewer #1: Yes

Reviewer #2: Yes

5. Review Comments to the Author

Reviewer #1: This is an excellent manuscript that advances the work on reliability and validity of rater evaluation methods. It is a creative application of a standardized methodology to an important clinical process, one that is broadly utilized throughout the world. The methods are sound, the results and analysis are clear, and the authors have done an excellent job considering and accounting for any potential pitfalls and confounds. It would be interesting to augment the manuscript to clarify how this method might or might not improve clinical practice down the line, whether there is any implication for other branches of clinimetrics, and whether this approach is applicable to evaluations of patient treatment in routine care.

Overall, this is an interesting analysis and an important application of rigorous analytic techniques to improve medical education and clinical training. I look forward to its publication and to potential dissemination across different domains.

Reviewer #2: Please discuss the applicability of the models discussed.

Please Simplify your results if possible in order to be understandable for all readers.

Simplify your conclusion to make it clear about the comparison of the rating models and the assumed new models.

6. PLOS authors have the option to publish the peer review history of their article (what does this mean?). If published, this will include your full peer review and any attached files.

Reviewer #1: No

Reviewer #2: **Yes: **Ghada Omer Hamad Abd El-Raheem

---

## [Author Response · Author response to Decision Letter 0]

16 Jun 2024

Thank you for considering our manuscript. We have revised the manuscript in accordance with the comments raised by the two reviewers. Our point-by-point responses to these comments are presented in a separate file labeled 'Response to Reviewers.' Newly added or revised text is indicated in blue font in the revised manuscript with tracked changes.

---

## [Decision Letter · Decision Letter 1]

21 Aug 2024

Item response theory model highlighting rating scale of a rubric and rater-rubric interaction in objective structured clinical examination

PONE-D-24-02840R1

Dear Dr. Uto,

We’re pleased to inform you that your manuscript has been judged scientifically suitable for publication and will be formally accepted for publication once it meets all outstanding technical requirements.

Kind regards,

Leona Cilar Budler

Academic Editor

PLOS ONE

Additional Editor Comments (optional):

No further comments

Reviewers' comments:

Reviewer's Responses to Questions

**Comments to the Author**

1. If the authors have adequately addressed your comments raised in a previous round of review and you feel that this manuscript is now acceptable for publication, you may indicate that here to bypass the “Comments to the Author” section, enter your conflict of interest statement in the “Confidential to Editor” section, and submit your "Accept" recommendation.

Reviewer #2: All comments have been addressed

2. Is the manuscript technically sound, and do the data support the conclusions?

Reviewer #2: Yes

3. Has the statistical analysis been performed appropriately and rigorously? 

Reviewer #2: Yes

4. Have the authors made all data underlying the findings in their manuscript fully available?

Reviewer #2: Yes

5. Is the manuscript presented in an intelligible fashion and written in standard English?

Reviewer #2: Yes

6. Review Comments to the Author

Reviewer #2: No other comments. The authors fulfilled the all the points. The manuscript is written in sound English and the results were meeting the objecyives of the study.

7. PLOS authors have the option to publish the peer review history of their article (what does this mean?). If published, this will include your full peer review and any attached files.

Reviewer #2: **Yes: **Ghada Omer Hamad Abd El-Raheem

---

## [Editor Report · Acceptance letter]

23 Aug 2024

PONE-D-24-02840R1 

PLOS ONE

Dear Dr. Uto, 

I'm pleased to inform you that your manuscript has been deemed suitable for publication in PLOS ONE. Congratulations! Your manuscript is now being handed over to our production team.

Kind regards, 

on behalf of

Dr. Leona Cilar Budler 

Academic Editor

PLOS ONE